# Patterns and determinants of incident cataract surgery in China from 2011 to 2015 using a nationally representative longitudinal database

Catherine Jan 🄳 ,[1,2,3] Xin Jin 🄳 ,[4,5] Yanhui Dong,[6] Thomas Butt,[7] Robert Chang,[8] Lisa Keay,[9] Mingguang He,[3,10] David Friedman,[11,12] Nathan Congdon[10,13,14]

For numbered affiliations see end of article.

**Correspondence to**
Dr Catherine Jan;
jlingxue@gmail.com and
Professor Nathan Congdon;
n.congdon@qub.ac.uk

## ABSTRACT

**Objectives** To investigate incident cataract surgery and to investigate determinants of cataract surgery uptake in Chinese adults.

**Design** This nationally representative longitudinal study recorded self-reported incident cataract surgery, and measured biological, clinical and socioeconomical characteristics at baseline and endline.

**Setting** In the first stage, 150 county-level units were randomly chosen with a probability-proportional-to-size sampling technique from a sampling frame containing all county-level units. The sample was stratified by region and within region by urban district or rural county and per capita gross domestic product. The final sample of 150 counties fell within 28 provinces of China.

**Participants** Urban and rural Chinese persons aged 45 years and older.

**Primary and secondary outcome measures** Incident cataract surgery (primary outcome) and the factors associated with incident cataract surgery (secondary outcome).

**Results** Among 16 663 people enrolled in 2011, 13 705 (82.2%) attended follow-up in 2015. Among these, 167 (1.22%) reported incident cataract surgery. Those receiving surgery were significantly older (66.2±8.79 vs 58.3±9.18, p≤0.001) and more likely to report: illiteracy (44.9% vs 27.1%, p<0.001), poor baseline distance vision (49.7% vs 20.0%, p≤0.001), poor baseline near vision (37.1% vs 21.8%, p≤0.001), baseline visual impairment (15.6% vs 5.5%, p≤0.001), diabetes (12.0% vs 7.42%, p≤0.05) and higher baseline depression scores (9.7 vs 8.4 on a scale of 0–30, p≤0.05). In linear regression models, older age, worse distance vision, hypertension or diabetes, illiteracy and lower depression score were significantly associated with undergoing surgery. Results were similar in models including only persons aged ≥60 years, except that urban residence was also associated with surgery. When only those aged ≥60 years with poor vision were included, results were again the same, except that higher household expenditure was also associated with surgery.

**Conclusions** In China, cataract surgical rates remain low; underserved groups such as rural dwellers are less likely to receive cataract surgery.

## STRENGTHS AND LIMITATIONS OF THIS STUDY

⇒ The strengths of this study include a large nationally representative sample from most provinces of China, selected using standardised protocols.

⇒ Study personnel were trained in standard fashion to ensure all procedures were carried out as described.

⇒ Our findings provide a basis for comparison with other low/middle-income countries and especially those undergoing rapid economic transitions.

⇒ Limitations include that the study did not measure objective visual acuity due to time and cost.

⇒ Another limitation is that our main outcome, having undergone cataract surgery, depended on patient report for similar reasons, though evidence has suggested that self-reported cataract surgery is accurate and reliable.

## INTRODUCTION

Blindness is among the most feared health conditions, ranking only after cancer and AIDS, and vision is the most valued of all the senses.[1] Age-related cataract is the world's and China's leading cause of blindness.[2 3] Surgery is currently the only available treatment for cataract, and has been rated among the most cost-effective procedures in medicine by the WHO.[4] Cataract surgery improves quality of life, increases activities of daily living and results in higher household economic productivity.[5 6] With a rapidly ageing population in China,[7] a significant increase in the burden of cataract is expected, highlighting the importance of efforts to address this burden.

Previous population-based, cross-sectional studies have reported the rates of prior cataract surgery in different regions of China,[8–12] with population 5-year incidence ranging from 2.9% in the Beijing Eye Study to 4.4% in Liwan, Southern China. Rural cataract surgical coverage (proportion of individuals with vision-impairing cataract who have received surgery) for those with vision of <6/60 rose from 35%

(range 24%–62%) in 2006[8] to 63% (range 43%–84%) in 2014, though coverage varied widely between provinces and was generally lower in poorer, western areas.[13] National prevalence and incidence data across China are vital for effective policy formulation and resource allocation. However, there have been no published studies of cataract surgery incidence using nationally representative samples in China[14 15] and most reports are from high-income countries.[16–18]

The multiple barriers to accessing cataract surgery in China include cost (direct out-of-pocket surgical expense as well as transportation and accommodation[19 20]), lack of knowledge about cataract and cataract surgery,[20 21] and concerns about the quality of local surgeons.[22] Prior work in this area has several shortcomings. First, no previous study has used a nationally representative sample which is needed because in a large country like China, social economic status and other determinants of cataract surgery uptake may vary widely by region. Second, many existing studies[19 20 22] have used data from prior to 2009, when China announced systematic health reforms. Access to cataract surgery in urban and rural areas has improved substantially due to universal coverage under the Urban Resident Medical Insurance and New Cooperative Medical System (NCMS) schemes.[23] Few recent publications using national data on cataract surgery have been published.[23] Finally, while evidence from low/middle-income countries suggests that health decisions are usually made at the household level, most existing studies in China have included only patient-level data, mostly concerning clinical rather than socioeconomic variables.[19]

To address these limitations, the present paper employed a national longitudinal survey conducted in 30 out of 31 mainland Chinese provinces (except Tibet, though the study did include one Tibetan county), representative of the adult population 45 years and older, in order to understand patterns of cataract surgery incidence across China between 2011 and 2015. The aim was to investigate incident cataract surgery, and to investigate the clinical (such as the presence of hypertension, diabetes and depression), biological and socioeconomic determinants of and barriers to cataract surgery in adults aged 45 years and above in urban and rural China between 2011 and 2015.

## METHODS
### Study population
The China Health and Retirement Longitudinal Study (CHARLS) is a nationally representative longitudinal survey among Chinese persons aged 45 years and older, which includes assessments of biological, social and economic conditions. We used data from the baseline study conducted between July and September 2011 ('wave 1') and a follow-up conducted (both surveys conducted in person) between July and September 2015 ('wave 3'). Main respondents and spouses in the baseline survey are followed throughout the life of CHARLS, or until they die.[24] Detailed information on the methodology of CHARLS has been described elsewhere.[24] In brief, 17 708 people aged 45 years and older participated

in the baseline study in 2011. Among them, 16 663 (94.1%) were eligible for data collection, 107 (0.604%) did not provide information on cataract surgery, 457 (2.58%) were aged below 45 years and 481 (2.72%) had already undergone cataract surgery before the baseline and were excluded from our sample. Among those eligible at baseline who provided information on cataract surgery in 2011 (n=16 991, 96.0% of the total), 13 705 (82.2%) attended the 4-year follow-up in 2015. The protocols of the follow-up for the variables used in this paper followed those of the baseline study.

Samples were chosen through multistage probability sampling. In the first stage, 150 county-level units were randomly chosen with a probability-proportional-to-size (PPS) sampling technique from a sampling frame containing all county-level units with the exception of the Tibet Autonomous Region. The sample was stratified by region and within region by urban district or rural county and per capita statistics on gross domestic product (GDP). The final sample of 150 counties fell within 28 provinces. Our sample used the lowest level of government organisation, consisting of administrative villages (*cun*) in rural areas and neighbourhoods (*shequ* or *juweihui*) in urban areas, as primary sampling units (PSUs). We selected three PSUs within each county-level unit, using PPS sampling (more details are reported elsewhere).[24]

### Variables
Cataract surgery incidence refers to the proportion of the cohort population aged 45 years and older who underwent cataract surgery in at least one eye between 2011 and 2015, among those who had not undergone cataract surgery in either eye at baseline. This was determined by participants' response to the question 'Have you had cataract surgery before?', comparing answers between the previous and present round.

Age and gender were self-reported variables taken from the baseline 2011 survey. Distance and near vision were self-reported at baseline, using a scale of 1–5, with 1 being excellent and 4 and 5 being fair and poor, respectively. Binary visual impairment was also self-reported.

Rural or urban place of residence was obtained from self-reported *hukou*. *Hukou* is a Chinese internal passport system that is designed to regulate the influx of rural residents into urban areas, and migrants with rural *hukou* do not have equal access to healthcare compared with local urban residents. *Hukou* also limits the type of public health insurance the person has access to.

Three comorbidities linked with cataract and/or uptake of cataract surgery were extracted from CHARLS. Hypertension was defined as either self-report of a physician's diagnosis or presence of objective biomarkers (systolic pressure ≥140 and/or diastolic pressure ≥90 mm Hg) during direct examination. Diabetes was similarly defined by self-report of prior diagnosis by a doctor or the presence of a biomarker (HbA1c value ≥6.5%) on examination. Depression was measured using the Center for Epidemiologic Studies Depression Scale[25] with total score ranging from 0 (minimal symptoms) to 30 (most symptomatic).

**Table 1** Baseline characteristics of participants aged 45 years and above who participated both in the baseline and follow-up surveys

| Baseline characteristic N=13705, ≥45, drop missing cases | Cataract surgery N=167 | No cataract surgery N=13538 |
|---|---|---|
| Age (years), mean (SD) | 66.17 (8.79) | 58.29 (9.18) |
| Distance vision status, n (%) | | |
| Poor or fair | 125 (75.9) | 7782 (57) |
| Excellent, very good and good | 32 (19.2) | 4904 (36) |
| Hypertension, n (%) | | |
| Yes | 67 (40) | 5238 (39) |
| No | 76 (46) | 6532 (48) |
| Diabetes, n (%) | | |
| Yes | 20 (12) | 1004 (7.4) |
| No | 102 (61) | 8701 (64) |
| Male | 71 (42) | 6556 (48) |
| Female gender, n (%) | 96 (58) | 6982 (52) |
| Literacy, n (%) | | |
| Literate | 92 (55) | 9865 (73) |
| Illiterate | 75 (45) | 3673 (27) |
| Mean depression score, mean (SD) | 9.70 (6.53) | 8.37 (6.30) |
| *Hukou*, n (%) | | |
| Rural | 127 (76.0) | 10961 (80) |
| Non-rural | 39 (23) | 2484 (18) |
| Log household per capita expenditure (US$), mean (SD) | 4.57 (0.94) | 4.68 (0.88) |
| Health insurance, n (%) | | |
| Yes | 158 (95) | 12664 (94) |
| No | 8 (5) | 797 (6) |
| Geographical location, n (%) | | |
| Eastern region | 52 (31) | 4704 (35) |
| Central region | 60 (36) | 4428 (33) |
| Western region | 55 (33) | 3306 (34) |
| Marital status | | |
| Married | 131 (78) | 11066 (82) |
| Not married | 36 (22) | 2472 (18) |

Data were obtained from the National School of Development (China Centre for Economic Research) of Peking University. GDP data were extracted from the China Statistical Yearbook, and per capita household expenditure was derived from surveys of household expenditure divided by the number of household members. GDP in Chinese yuan was converted to the US dollar using the exchange rate on 1 August 2011, released by the centre for foreign trade, Bank of China.

All data collected in the CHARLS are maintained at the National School of Development of Peking University, Beijing, China. The datasets are available from https://charls.charlsdata.com/pages/data/111/zh-cn.html.[26]

## Patient and public involvement

As this is an analysis of publicly available dataset, neither ethics nor patient and public involvement is applicable.

## Statistical analysis

All statistical analyses were performed using Stata V.13.0 (StataCorp, College Station, Texas, USA). The t-test was used to compare continuous variables, while Pearson $X^2$ or Fisher's exact tests were used for the comparison of categorical data. As a test to assess accuracy of household expenditure data from the sample as an index of income in the province, a scatterplot graph was made comparing household expenditure with provincial GDP data. The 4-year incidence of cataract surgery stratified by age and gender was calculated. Age was defined in our analysis as the age at baseline. Univariate and multivariate logistic regression models were used to test for association with incident cataract surgery over the 4-year follow-up period. We ran three sets of regression analysis, one including the whole sample of eligible participants (ie, all eligible people who were aged 45 years and above), one including those eligible participants aged 60 years and above (as visually significant cataract is far more prevalent in this age range), and the third including those eligible participants aged 60 years and above who also reported fair or poor vision (the subset of participants at greatest risk of undergoing cataract surgery). The variables in univariate models that had p value of less than 0.05 were deemed significant and were kept in the multivariate models. The incidence of cataract surgery in each province was affected by the age and gender structure of the population aged 45 years and over. Therefore, the provincial incidence of cataract surgery was adjusted for age and gender structure of the population. We applied sample weights to the cataract surgery incidence between 2011 and 2015. These weights were constructed from weights for the structure of age and gender from the baseline wave of the CHARLS. A p value of <0.05 was defined as statistically significant.

## RESULTS

Among the 16663 eligible people aged 45 years and above (table 1), 13705 (82.2%) attended the year 4 follow-up. Among them, 13538 (98.8%) did not report receiving surgery between 2011 and 2015, while 167 (1.22%) did. Those who received surgery were significantly older (66.17±8.79 vs 58.29±9.18, p≤0.001), and more likely to be illiterate (45% vs 27%, p<0.001), have self-reported poor distance vision at baseline (49.7% vs 20.0%, p≤0.001), self-reported poor near vision at baseline (37.1% vs 21.8%, p≤0.001), self-reported visual impairment at baseline (15.6% vs 5.5%, p≤0.001), diabetes (12.0% vs 7.4%, p≤0.05) and higher depression scores at baseline (9.70 vs 8.37 on a scale of 0–30, p≤0.05).

Table 2 lists the 28 provinces for which data are available in the current study in order of per capita GDP by region and by province within region. There was a nonsignificant trend towards higher mean weekly household per capita expenditure in Eastern China (US$34.3) versus

**Table 2** Characteristics of 28 Chinese provinces with regard to per capita gross domestic product (GDP), population and other characteristics, in 2011 (arranged by region and province in order of reported per capita GDP)

| Province name | Reported per capita GDP (US$, 2011) | Reported total population (thousands) (2011) | Reported population density (persons/km²) (2011) | Number of participants in the current study (N=16663) | Mean weekly household per capita expenditure of study participants (n=16663), 2011 (US$) | Mean household per capita expenditure, 2011, by geographical region (US$) |
|---|---|---|---|---|---|---|
| Eastern China | | | | | | |
| Tianjin | 13232.0 | 13550 | 2636 | 149 | 44.9 | 34.3 |
| Beijing | 12680.0 | 20190 | 1428 | 85 | 73.8 | |
| Shanghai | 12820.1 | 23470 | 3702 | 58 | 58.6 | |
| Jiangsu | 9672.5 | 78990 | 2013 | 834 | 38.8 | |
| Zhejiang | 9200.3 | 54630 | 1741 | 659 | 48.0 | |
| Guangdong | 7889.4 | 105050 | 2637 | 913 | 31.9 | |
| Liaoning | 7882.1 | 43830 | 1712 | 510 | 24.3 | |
| Fujian | 7356.8 | 37200 | 2306 | 483 | 36.5 | |
| Shandong | 7350.3 | 96370 | 1389 | 1374 | 24.4 | |
| Jilin | 5972.1 | 27490 | 2371 | 415 | 32.6 | |
| Hebei | 5274.8 | 72410 | 2362 | 712 | 24.1 | |
| Heilongjiang | 5096.2 | 38340 | 5146 | 312 | 39.5 | |
| Total | 8702.2 | 50960 | 2454 | 6504 | 39.8 | |
| Central China | | | | | | |
| Hubei | 5310.2 | 57580 | 1969 | 518 | 27.7 | 20.5 |
| Shanxi | 4869.2 | 35930 | 2977 | 511 | 22.3 | |
| Hunan | 4639.8 | 65960 | 2908 | 773 | 28.3 | |
| Henan | 4450.5 | 93880 | 5124 | 1313 | 24.7 | |
| Jiangxi | 4060.6 | 44880 | 4527 | 815 | 29.7 | |
| Anhui | 3984.4 | 59680 | 2265 | 816 | 23.4 | |
| Total | 4552.5 | 59652 | 3295 | 4746 | 26.0 | |
| Western China | | | | | | |
| Inner Mongolia | 9002.3 | 24820 | 764 | 819 | 44.3 | 18.1 |
| Chongqing | 5357.2 | 29190 | 1830 | 219 | 29.5 | |
| Shaanxi | 5196.4 | 37430 | 5821 | 532 | 45.3 | |
| Xinjiang | 4672.0 | 22090 | 4563 | 96 | 26.2 | |
| Qinghai | 4584.2 | 5680 | 2487 | 144 | 25.1 | |
| Sichuan | 4058.0 | 80500 | 2782 | 1433 | 25.7 | |
| Guangxi | 3932.7 | 46450 | 1569 | 582 | 22.2 | |
| Gansu | 3042.7 | 25640 | 3824 | 402 | 24.8 | |
| Yunnan | 2991.5 | 46310 | 3811 | 997 | 21.7 | |
| Guizhou | 2548.6 | 34690 | 3502 | 189 | 23.9 | |
| Total | 4538.6 | 35280 | 3095 | 5413 | 28.9 | |

Central (US$20.5) and Western China (US$18.1). Mean provincial per capita weekly household expenditures as reported by study participants were highly correlated ($R^2$=0.650, p=0.0018, regression correlation) with available data for per capita GDP by province (figure 1).

Among 7052 (42.3%) participants in the current study aged 60 years and above, 126 indicated they had undergone cataract surgery during 2011–2015. Among 4522 (27.1%) participants aged 60 years and above and reporting 'fair' or 'poor' vision at the time of the most recent survey 4 years previously, data on cataract surgery were missing for 38 persons

(0.840%), and a total of 98 (2.19% or 98 of 4484) indicated they had undergone cataract surgery, the 4-year incidence rate ranging from 2.43% (31 of 1401) in Central China, to 2.27% (35 of 1545) in the East to 2.08% (32 of 1538) in the West (table 3).

In linear regression models of potential predictors of regional incidence of cataract surgery among persons aged 45 years and above at baseline, older age, worse vision, having hypertension or diabetes, being illiterate and higher depression score were all significantly associated with increased surgery rates in the univariate model, and all these factors

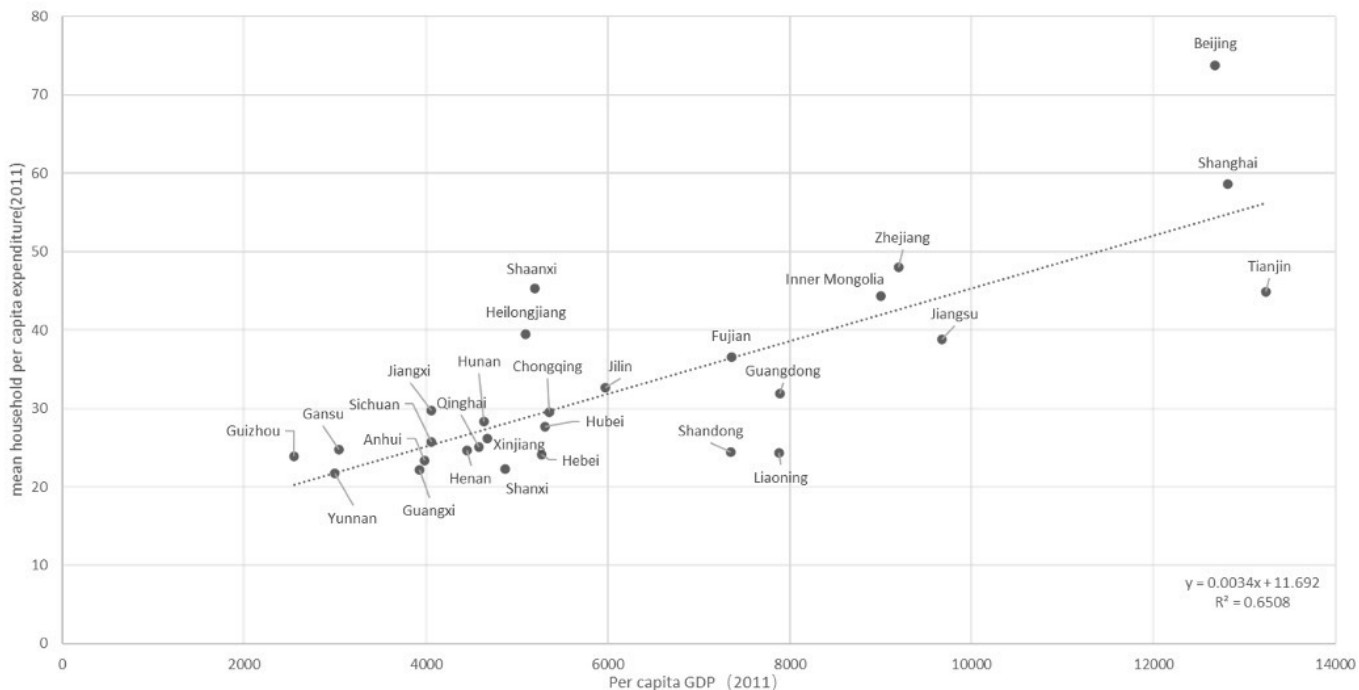

**Figure 1** Scatterplot and regression line showing the reported per capita gross domestic product (GDP) of provinces (US$) against the mean household per capita expenditure (US$) of participants in our study from these provinces.

remained significant in multivariate model (table 4). Results were similar in the multivariate regression model including only persons aged 60 years and above, except that urban *hukou* was also associated with higher incidence of surgery (table 5). When only those aged 60 years and above with self-reported fair or poor vision were included, results in the multivariate model were again the same, except that higher household expenditure was also associated with higher incidence of surgery (table 6).

## DISCUSSION

The current study found that between 2011 and 2015, the 4-year incidence of self-reported cataract surgery among people aged 45 years and above was relatively low (1.22%) compared with earlier reports, which ranged from 2.9% (5-year incidence) in the Beijing Eye Study[10] to 4.4% (5-year incidence) in Liwan, Southern China.[12] It is worth noting that the Beijing and Liwan studies reported 5-year incidence rates, whereas our study reported 4-year incidence rate. This discrepancy may partly explain why our incidence rate appears to be lower. However, due to study limitations, we were unable to extend the follow-up period to 5 years. It is important to note that our sample was a nationally representative randomised sample selected from all provinces of China, from both urban and rural regions in each province, while previous studies

**Table 3** Characteristics of persons aged 60 years and over with self-reported vision of fair or poor in 2011 (N=4522) by region

| Economic region | Mean age (SD) | Women, n (%) | Literate, n (%)* | Mean depression score (95% CI) (SD)*** | Rural (vs urban) residence, n (%)* | Mean household per capita expenditure, US$ (SD) | Has only public health insurance, n (%)*** | 4-year cataract surgery incidence*, (%) |
|---|---|---|---|---|---|---|---|---|
| Eastern region | 68.7 (0.181) | 814 (52.7) | 978 (63.3) | 9.18 (8.86 to 9.50) (0.164) | 1151 (74.6) | 24.7 (1.05) | 1399 (90.6) | 35/1,545 (2.27) |
| Central region | 67.8 (0.177) | 705 (50.3) | 840 (60.0) | 10.1 (9.78 to 10.49) (0.179) | 1130 (80.7) | 19.7 (0.728) | 1292 (92.2) | 31/1,401 (2.43) |
| Western region | 68.4 (0.170) | 791 (51.4) | 909 (59.1) | 10.9 (10.57 to 11.24) (0.17) | 1228 (79.8) | 21.2 (0.854) | 1397 (90.8) | 32/1,538 (2.08) |
| Total | 68.34 (0.101) | 2327 (51.46) | 2727/4484 (60.82) | 10.07 (9.88 to 10.23) (0.099) | 3509/4482 (78.29) | 21.8 (0.5) | 4088/4422 (92.45) | 98/4484 (2.19) |

Regions are listed in order of per capita GDP in 2011.
*P<0.05, **p<0.01, ***p<0.001, $X^2$ test for trend from highest to lowest cataract surgery incidence.
*Data on cataract surgery were missing for 38 persons (0.840%).
GDP, gross domestic product.

**Table 4** Linear regression model of potential predictors of the incidence of cataract surgery among all persons aged ≥45 years (n=16 663, among whom 167 (1.00%) reported having surgery) across whole of China between 2011 and 2015

| Baseline characteristic | Univariate model | | | Multivariate model | | |
|---|---|---|---|---|---|---|
| | Beta coefficient | 95% CI | P value | Beta coefficient | 95% CI | P value |
| Age | 0.094 | **−0.0013 to −0.0009** | <0.001 | 0.080 | **0.00098 to 0.00106** | <0.001 |
| Poor or fair self-reported vision | **0.41** | **0.005 to 0.013** | <0.001 | 0.022 | **0.00485 to 0.00625** | <0.001 |
| Hypertension present (adjusted for age) | 0.005 | **0.001 to 0.002** | <0.001 | −0.018 | **−0.0048 to −0.0035** | <0.001 |
| Diabetes present (adjusted for age) | 0.023 | **0.008 to 0.010** | <0.001 | 0.019 | **0.0063 to 0.0085** | <0.001 |
| Female sex | 0.013 | −0.001 to 0.007 | 0.129 | | | |
| Literate | −0.0438 | **−0.015 to 0.007** | <0.001 | −0.021 | **−0.00614 to −0.00468** | <0.001 |
| Mean depression score (adjusted for age) | 0.023 | **0.0004 to 0.0005** | <0.001 | 0.012 | **0.00017 to 0.00028** | <0.001 |
| Rural (vs non-rural) *hukou* | −0.014 | −0.0008 to 0.0085 | 0.106 | | | |
| Log household per capita expenditure | −0.013 | −0.0038 to 0.0006 | 0.146 | | | |
| Reported having health insurance | 0.005 | −0.0054 to 0.0102 | 0.550 | | | |

Bold type indicates values significant at p<0.05.

only reported findings from one specific urban region in a particular province. This difference in sample selection may also contribute to the variance in incidence rates. Moreover, in our study, cataract surgery was self-reported, whereas in other studies, it was extracted from medical records or clinical examination. More data are needed to help policymakers address the barriers to uptake of cataract surgery in China.[19–22] Previous studies have examined institutional predictors of higher surgical capacity at the institutional level (principally screening outreach activities). This is the first large population study of determinants of surgery uptake in China, providing complementary information.

Our finding of lower surgical rates in more rural provinces is in agreement with results from prior population studies,[8 13] which have also reported especially low rates of surgical uptake in rural settings. This finding indicates

that this lower rate persists despite the rapid expansion of the NCMS rural health insurance programme.[27 28]

With respect to individual-level variables, it was unsurprising that worse self-reported vision, as captured by three different variables, was significantly associated with greater uptake of surgery. As expected, older age was associated with elevated uptake of surgery.

Diabetes is well established in population studies to be associated with higher risk of lens opacity, and so our observed higher prevalence of diabetes among those accepting surgery is not surprising. Additionally, cataract has been linked to higher rates of mental health problems, including depression,[29] which we believe could account for the observed higher rates of depression among those undergoing incident cataract surgery. To our knowledge, there is no evidence suggesting that depression drugs cause cataract. We conducted a thorough search of

**Table 5** Linear regression model of potential predictors of the regional incidence of cataract surgery among all persons aged ≥60 years (n=7052, among whom 126 (1.79%) reported having surgery)

| Baseline characteristic | Univariate model | | | Multivariate model* | | |
|---|---|---|---|---|---|---|
| | Beta coefficient | 95% CI | P value | Beta coefficient | 95% CI | P value |
| Age | **0.060** | **0.0007 to 0.020** | <0.001 | 0.036 | **0.00077 to 0.00096** | <0.001 |
| Poor or fair self-reported vision | 0.052 | **−0.025 to −0.008** | <0.001 | 0.039 | **0.01167 to 0.01431** | <0.001 |
| Hypertension present (adjusted for age) | −0.011 | **−0.0043 to −0.0023** | <0.001 | −0.023 | **−0.00804 to −0.00567** | <0.001 |
| Diabetes present (adjusted for age) | 0.029 | **0.0117 to 0.0151** | <0.001 | 0.024 | **0.00949 to 0.01318** | <0.001 |
| Female sex | 0.024 | −0.00005 to 0.01499 | 0.068 | | | |
| Literate | −0.032 | **−0.0178 to 0.0017** | **0.018** | −0.027 | **−0.00957 to −0.00701** | <0.001 |
| Mean depression score (adjusted for age) | 0.020 | **0.000383 to −0.000536** | <0.001 | 0.023 | **0.00044 to 0.00062** | <0.001 |
| Rural (vs non-rural) *hukou* | −0.030 | **−0.0204 to −0.0015** | **0.024** | −0.043 | **−0.01801 to −0.01489** | <0.001 |
| Log household per capita expenditure | 0.011 | −0.00277 to 0.00642 | 0.435 | | | |
| Reported having health insurance | 0.001 | −0.016 to 0.017 | 0.926 | | | |

Bold type indicates values significant at p<0.05.
*Variables that showed significance in the univariate model were included in the multivariate model.

**Table 6** Linear regression model of potential predictors of the regional incidence of cataract surgery among all persons aged ≥60 years and those who reported fair or poor vision (n=4522)

| Baseline characteristic | Univariate model | | | Multivariate model* | | |
|---|---|---|---|---|---|---|
| | Beta coefficient | 95% CI | P value | Beta coefficient | 95% CI | P value |
| Age | 0.064 | 0.0008227 to 0.002492 | <0.001 | 0.035 | 0.00076 to 0.00101 | <0.001 |
| Hypertension present (adjusted for age) | −0.014 | −0.00596 to −0.0032489 | <0.001 | −0.039 | −0.01425 to −0.01100 | <0.001 |
| Diabetes present (adjusted for age) | 0.041 | 0.0184 to 0.0229 | <0.001 | 0.041 | 0.01744 to 0.02250 | <0.001 |
| Female sex | 0.024 | −0.00278 to 0.01839 | 0.148 | | | |
| Literate | −0.027 | −0.01995 to 0.00171 | 0.099 | | | |
| Mean depression score (adjusted for age) | 0.006 | 0.0000485 to 0.0002474 | 0.004 | 0.027 | 0.00053 to 0.00078 | 0.004 |
| Rural (vs non-rural) *hukou* | −0.041 | −0.030558 to −0.003757 | 0.012 | −0.031 | −0.01515 to −0.01066 | <0.001 |
| Log household per capita expenditure | 0.036 | 0.0001394 to 0.012749 | 0.045 | 0.012 | 0.00130 to 0.0032 | <0.001 |
| Reported having health insurance | −0.001 | −0.022986 to 0.021785 | 0.958 | | | |

Bold type indicates values significant at p<0.05.
*Variables that showed significance in the univariate model were included in the multivariate model.

PubMed and were unable to find any studies supporting a causal link between depression drugs and cataract formation. A noteworthy individual-level finding was a higher prevalence of self-reported illiteracy among those who underwent incident cataract surgery. Previous population studies have consistently linked poor education and lack of knowledge about cataract with low surgical uptake.[22 23] One possible explanation for this finding is that illiterate people were more likely to comply with the doctor's recommendations. Conversely, those with higher levels of education may have been more cautious about using local services and may have been unable to afford distant care. Additionally, those with lower literacy levels may have other unmeasured factors that predispose them to higher rates of cataract, such as greater exposure to ultraviolet light. It is also possible that illiterate individuals provided inaccurate information regarding their cataract surgery or had other eye-related procedures, which resulted in relatively unreliable data concerning cataract surgery.

Higher economic status (represented by higher weekly household per capita expenditure, which correlated strongly with per capita GDP, figure 1) was associated with receiving cataract surgery among people aged 60 years and above with poor vision. This result is expected; however, there have been several efforts in recent years that help provide surgery to those economically more disadvantaged. One is the roll-out of projects providing high-volume free surgery, such as the Million People Cataract Surgery Plan,[30] during the period covered in the current study. This has been supplemented by non-governmental organisation-supported cataract surgical programmes by entities such as Lifeline Express,[31] and by the high penetration of the NCMS rural health insurance system, now covering some 98.9% of rural dwellers[27] and substantially reducing out-of-pocket expenses associated with cataract surgery. We observed that the percentages of people covered by public health insurance did not vary significantly between poorer and richer regions. These programmes taken together may have improved access in poor areas that reduced inequality in cataract surgery access associated with income, but in the end, those with greater income were more likely to have cataract surgery.

The strengths of this study include a large nationally representative sample from most provinces of China, selected using standardised protocols. Study personnel were trained in standard fashion to ensure all procedures were carried out as described. Our findings provide a basis for comparison with other low/middle-income countries and especially those undergoing rapid economic transitions. Limitations must be acknowledged as well. In the first place, the study did not measure objective visual acuity due to time and cost. Our main outcome, having undergone cataract surgery, depended on patient report for similar reasons, though evidence has suggested that self-reported cataract surgery is accurate and reliable.[32] Finally, the results may be influenced by patients (17.8%) who did not come for the follow-up examination.

## Conclusion

Our study provides unique data on cataract surgical incidence in China. Our results suggest that surgical rates remain low. Both rural and metropolitan have access to cataract surgery, but underserved groups such as rural dwellers are less likely to receive cataract surgery.

**Author affiliations**
[1]Department of Ophthalmology and Surgery, Faculty of Medicine, Dentistry & Health Sciences, The University of Melbourne, East Melbourne, Victoria, Australia
[2]Lost Child's Vision Project, Sydney, New South Wales, Australia
[3]Centre for Eye Research Australia, Royal Victorian Eye and Ear Hospital, East Melbourne, Melbourne, Australia
[4]Institute of Population Research, Peking University, Beijing, People's Republic of China
[5]Peking University Library, Peking University, Beijing, People's Republic of China
[6]Institute of Child and Adolescent Health, School of Public Health, Peking University, Beijing, People's Republic of China
[7]UCL Institute of Ophthalmology, University College London, London, UK
[8]Byers Eye Institution, Stanford University, Palo Alto, California, USA
[9]School of Optometry and Vision Science, University of New South Wales, Sydney, New South Wales, Australia

[10]Zhongshan Ophthalmic Center, Sun Yat-Sen University, Guangzhou, China
[11]Department of Ophthalmology, Harvard University, Boston, Massachusetts, USA
[12]Massachusetts Eye and Ear Infirmary, Boston, Massachusetts, USA
[13]TREE Centre, Centre for Public Health, Queen's University Belfast, Belfast, UK
[14]ORBIS International, New York, New York, USA

**Acknowledgements** The authors would like to thank Professor Zhao Yaohui for her valuable comments and suggestions on this manuscript. They would also like to thank the CHARLS team for the data that enabled these analyses. This research is supported by the Wellcome funding (R2806CPH).

**Contributors** CJ conceptualised the study, did a literature review, participated in data collection and data analysis, drafted the manuscript and is responsible for the overall content as a guarantor. NC conceptualised the study, interpreted data and revised the manuscript. JX participated in data collection and data analysis, interpreted data, created the figure and revised the manuscript. DF, YD, MH, RC, TB and LK participated in data interpretation and manuscript revision. All authors read and approved the final draft.

**Funding** The authors have not declared a specific grant for this research from any funding agency in the public, commercial or not-for-profit sectors.

**Competing interests** None declared.

**Patient and public involvement** Patients and/or the public were not involved in the design, or conduct, or reporting, or dissemination plans of this research.

**Patient consent for publication** Not required.

**Ethics approval** This study involves human participants. The CHARLS was approved by the Ethical Review Committee of Peking University, and all participants gave written informed consent at the time of participation. This study used public data and therefore does not require additional ethics approval. The original CHARLS from which the data were collected was carried out in Beijing and has obtained ethics approval from the Ethical Review Committee of Beijing University; the ethical approval number is IRB00001052-11015.

**Provenance and peer review** Not commissioned; externally peer reviewed.

**Data availability statement** Data are available in a public, open access repository. All data collected in the CHARLS are maintained at the National School of Development of Peking University, Beijing, China. The datasets are available from http://charls.pku.edu.cn/pages/data/111/zh-cn.html.

**ORCID iDs**
Catherine Jan http://orcid.org/0000-0001-7383-8208
Xin Jin http://orcid.org/0000-0001-8712-4513

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
