## [Reviewer comments · BMJ Open]

ARTICLE DETAILS

TITLE (PROVISIONAL)	Patterns and Determinants of Incident Cataract Surgery in China from 2011 to 2015 Using a Nationally-representative Longitudinal Database
AUTHORS	Jan, Catherine; Xin, Jin; Dong, Yanhui; Butt, Thomas; Chang, Robert; Keay, Lisa; He, Mingguang; Friedman, David; Congdon, Nathan

VERSION 1 – REVIEW

REVIEWER	Richard Hida Universidade de Sao Paulo Hospital das Clinicas
REVIEW RETURNED	16-Jan-2023

GENERAL COMMENTS	Dear authors, BMJ Open Manuscript Number: bmjopen-2022-069702 Title: Patterns and Determinants of Incident Cataract Surgery in China from 2011 to 2015 using a Nationally-representative Longitudinal Database. GENERAL CONSIDERATIONS - The authors have studied the subpopulations at greatest risk for low utilization of cataract surgery, and to investigate determinants of cataract surgery in Chinese adults.- Minor native English correction must be performed in general.- Cataract surgery “utilization” is not a “risk”. Please review throughout the article.- If I may suggest other titles: “Patterns and Determinants of Incident Cataract Surgery (in China)” ABSTRACT - The authors have mentioned: “To identify subpopulations at greatest risk for low utilization of cataract surgery, and to investigate determinants of cataract surgery in Chinese adults.”- The authors have mentioned as conclusion: “In China, cataract surgical rates remain low, but it appears governmental and non-governmental programs have reduced income-related barriers to access.”- Please do not mention sentences that cannot be supported by data shown. If cataract surgical rates remain low or high is not part of the purpose of the study. If governmental programs have reduced income-related barriers for access is also not part of this study. Please delete this sentence and conclude something related to the purpose and data shown. RESULTS - Please organize what is MATERIAL (sample), METHOD (how
---

	study was performed), and RESULT (data analyzed). DISCUSSION  - The authors have mentioned as purpose: "The aim was to identify subpopulations at greatest risk for low utilization, and to investigate the clinical, biological, and socio-economic determinants of and barriers to cataract surgery in adults aged 45 years and above in urban and rural China between 2011 and 2015" - The authors have mentioned as conclusion: "Our study provides unique data on cataract surgical incidence in China. Our results suggest that rural and metropolitan both have access to cataract surgery. Surgical rates remain low, but reveal an encouraging phenomenon that a variety of governmental and non-governmental programs may have reduced income-related barriers to accessing cataract surgery in China." - Please do not conclude sentences that cannot be supported by data shown. - Cataract surgical incidence was never an issue here. Please review this sentence. - Please focus on identifying the subpopulation with low incidence of cataract surgery. Please also focus answering clinical, biological and socio-economic factors as mentioned in the purpose. - Surgical rates, governmental programs and access to cataract surgery are not part of this study (is it?). Please delete this sentence and conclude something.
--	--

REVIEWER	Jasmin Zvorničanin University Clinical Center Tuzla, Ophthalmology
REVIEW RETURNED	17-Jan-2023

GENERAL COMMENTS	This rigorous study addresses an important question regarding the determinants of cataract surgery in a Chinese adult population. The methodology used to create the study, primarily the selection of the sample, is clear, standardized and previously published. However, the results and conclusions of this research require a more detailed explanation. Abstract Methods - it is not clear from the methods presented in the paper which "clinical characteristics" were used. Does it refer to ophthalmic or systemic characteristics? Please explain. Results - the presented results of linear regression can be shortened and simplified to make them clearer to researchers who have less knowledge of statistical methods. Conclusion - the results presented in the research as well as in the abstract itself do not support the conclusion that the level of income represented a significant barrier to the use of cataracts. What this study adds? Are there previous studies on surgery uptake from China? References: 8-13, 19, 20, 22 and 23. Introduction The complete introduction is relatively long and more narrative in nature, in which a significant part of the focus on the basic task of this research is lost. Page 5, First paragraph, Line 5 - Sentence: "With a rapidly aging....." please provide an adequate reference or find one on a global scale. Page 5, Second paragraph, Line 6 - How did you get the information that India is a richer country?
---

	Page 5, Second paragraph, Line 10 - The sentence "The multiple barriers..." is almost identical in discussion. No need to repeat.... Page 6, First paragraph, Line 1 - Sentence "Access to cataract surgery..." how did you come to this conclusion The results Are the results influenced by the fact that a significant number of patients (17.8%) did not come for the follow-up examination. Explain this in the limitations of the study. Page 10, First paragraph - Please do not share the percentage values that are higher than those listed in Table 1. Table 1. Marital status may be shortened to: married or not Table 2. What is the significance of this Table? Table 3. What is the significance of this Table or is everything important already said in the paragraph above? Discussion Page 21, First paragraph - Why did you compare the results from the population over 60 years old in your research with the results of other studies that presented the results from the population over 45 or 50 years old? Are there any other studies from around the world that show the incidence of cataract surgery? Why are your results lower? Suggestion: does your study reflect real results while earlier results are actually limited? Page 22, Second paragraph - The results that people who are illiterate, and on the other hand that people with higher incomes have a higher frequency of cataract surgery, are disproportionate. In addition, take into account the smaller number of operations in rural areas. The next page states that better economic status has a higher incidence of cataracts. This needs clarification! Is there a chance that illiterate people gave wrong information about cataract surgery, and that they had other eye-related procedures? It can be concluded that people who have cataracts have a higher chance of having depression. Could it also be the reverse case - that some drugs accelerate the cataract development? In the discussion, it is mentioned in several places that the implemented programs have improved the situation related to the availability of cataract surgery. On what basis can we come to that conclusion, which comparison was made and with what results? On the other hand, the results of this research do not show that? Additionally, could you explain is there a sufficient number of doctors (we have data), is there enough equipment (perhaps you have data) or is the insufficient patient education the biggest problem? As a conclusion, are "great efforts" still needed to meet the needs for cataract surgery of the growing population? What should we pay attention to? References The cited references are adequate, with a note that there are a couple of other studies (although they are from other populations) that could be included in the discussion of this paper.
--	---

VERSION 1 – AUTHOR RESPONSE

Reviewer: 1

Dr. Richard Hida, Universidade de Sao Paulo Hospital das Clinicas, Universidade Federal de Sao Paulo Escola Paulista de Medicina

Comments to the Author:

Dear authors,

BMJ Open

Manuscript Number: bmjopen-2022-069702

Title: Patterns and Determinants of Incident Cataract Surgery in China from 2011 to 2015 using a Nationally-representative Longitudinal Database.

GENERAL CONSIDERATIONS

- The authors have studied the subpopulations at greatest risk for low utilization of cataract surgery, and to investigate determinants of cataract surgery in Chinese adults.

- - Minor native English correction must be performed in general.
- **Thank you for your comment. The language has been proof-read by six native speakers in the author list.**
- - Cataract surgery “utilization” is not a “risk”. Please review throughout the article.
- **Thank you for pointing that out. We have modified the language of our first aim to “To investigate incident cataract surgery.” We have also revised the entire manuscript and deleted surgery utilization.**
- - If I may suggest other titles: “Patterns and Determinants of Incident Cataract Surgery (in China)”
- **Thank you for your suggestion. After discussing with the author group, we decided to keep our original title as it gives a more holistic view of the article’s scope. Hope you understand.**

ABSTRACT

- The authors have mentioned: “To identify subpopulations at greatest risk for low utilization of cataract surgery, and to investigate determinants of cataract surgery in Chinese adults.”

- The authors have mentioned as conclusion: “In China, cataract surgical rates remain low, but it appears governmental and non-governmental programs have reduced income-related barriers to access.”

- Please do not mention sentences that cannot be supported by data shown. If cataract surgical rates remain low or high is not part of the purpose of the study. If governmental programs have reduced income-related barriers for access is also not part of this study. Please delete this sentence and conclude something related to the purpose and data shown.

Thank you for your helpful suggestion. We have modified the aims section so that now it reads: “To investigate incident cataract surgery, and to investigate determinants of cataract surgery uptake in Chinese adults.” (Page 3, line 4-5). We show in Table 6 that underserved groups such as rural dwellers are less likely to receive cataract surgery. Our conclusion relates to what we found with regards to sub-populations at greatest risk for low surgery incidence and determinants of uptake. Now the conclusion reads: “In China, cataract surgical rates remain low, underserved groups such as rural dwellers are less likely to receive cataract surgery.” (Page 4, lines 6-7)

RESULTS

- Please organize what is MATERIAL (sample), METHOD (how study was performed), and RESULT (data analyzed).

We have organised Material as follows: “Among the 16,663 eligible people aged 45 years and above (Table 1), 13,705 (82.2%) attended the year four follow-up. Among them, 13,538 (98.8%) did not report receiving surgery between 2011 and 2015, while 167 (1.22%) did.” (Lines 2-3, page 10)

We have organised Method in the Method section (Pages 6-9).

We have organised Result in our Result section (Pages 10-21). Statistical analysis was outlined in the Method section (Lines 6-23, page 9).

DISCUSSION

- The authors have mentioned as purpose: "The aim was to identify subpopulations at greatest risk for low utilization, and to investigate the clinical, biological, and socio-economic determinants of and barriers to cataract surgery in adults aged 45 years and above in urban and rural China between 2011 and 2015"

- The authors have mentioned as conclusion: "Our study provides unique data on cataract surgical incidence in China. Our results suggest that rural and metropolitan both have access to cataract surgery. Surgical rates remain low, but reveal an encouraging phenomenon that a variety of governmental and non-governmental programs may have reduced income-related barriers to accessing cataract surgery in China."

- - Please do not conclude sentences that cannot be supported by data shown.
- - Cataract surgical incidence was never an issue here. Please review this sentence.

- Please focus on identifying the subpopulation with low incidence of cataract surgery. Please also focus answering clinical, biological and socio-economic factors as mentioned in the purpose.

- Surgical rates, governmental programs and access to cataract surgery are not part of this study (is it?). Please delete this sentence and conclude something.

Thank you for your suggestion. We have deleted that sentence according to your suggestion. Instead, we talked about underserved groups such as rural dwellers are less likely to receive cataract surgery (Table 6). Now the conclusion reads like this: "Our study provides unique data on cataract surgical incidence in China. Our results suggest that surgical rates remain low. Rural and metropolitan both have access to cataract surgery, but underserved groups such as rural dwellers are less likely to receive cataract surgery." (Page 23, lines 11-13)

Reviewer: 2

Dr. Jasmin Zvorničanin, University Clinical Center Tuzla

Comments to the Author:

This rigorous study addresses an important question regarding the determinants of cataract surgery in a Chinese adult population. The methodology used to create the study, primarily the selection of the sample, is clear, standardized and previously published. However, the results and conclusions of this research require a more detailed explanation.

Thank you for your comment.

Abstract

Methods - it is not clear from the methods presented in the paper which "clinical characteristics" were used. Does it refer to ophthalmic or systemic characteristics? Please explain.

As stated under the Variable section in our Method, we have presented the following characteristics: Age, gender, distance and near vision, binary visual impairment, rural or urban place of residence, three comorbidities linked with cataract and/or uptake of cataract surgery: hypertension (clinically measured), diabetes (clinically measured) and depression (clinically measured) (Pages 7-8).

Results - the presented results of linear regression can be shortened and simplified to make them clearer to researchers who have less knowledge of statistical methods.

Thank you for your suggestion, however, the variables in our regression models have been carefully chosen by both ophthalmologists and public health scientists who believe that those are important variables to be included. Therefore, we would like to respectfully keep our linear regression models rather than shortening or simplifying them. If there is anything that is unclear in the model, we are more than happy to provide further explanation.

Conclusion - the results presented in the research as well as in the abstract itself do not support the conclusion that the level of income represented a significant barrier to the use of cataracts.

We would like to respectfully point out that we have indeed showed in Table 6 that lower income was associated with lower uptake of surgery. $P < 0.0001$ and the Beta is positive. Table 6 also showed that rural dwellers are less likely to receive cataract surgery.

What this study adds?

Are there previous studies on surgery uptake from China? References: 8-13, 19, 20, 22 and 23.

Yes, there have been previous studies on surgery uptake from China and elsewhere. The details have been reported in our Introduction section (lines 10-24 of page 5, and lines 1-6 of page 6).

Introduction

The complete introduction is relatively long and more narrative in nature, in which a significant part of the focus on the basic task of this research is lost.

Page 5, First paragraph, Line 5 - Sentence: "With a rapidly aging....." please provide an adequate reference or find one on a global scale.

Thank you for your suggestion. We have added the following reference: Lopreite M, Zhu Z. The effects of ageing population on health expenditure and economic growth in China: A Bayesian-VAR approach. Social science & medicine. 2020 Nov 1;265:113513.

Page 5, Second paragraph, Line 6 - How did you get the information that India is a richer country?

In our text we have stated that India is a less wealthy country. However, per your comment on the cumbersome introduction, we have deleted this sentence as it is less relevant to the aims of the study. (line 15, page 5)

Page 5, Second paragraph, Line 10 - The sentence "The multiple barriers..." is almost identical in discussion. No need to repeat....

Thank you for your helpful suggestion. We have deleted the repeated content in the discussion (Page 21, lines 7-8).

Page 6, First paragraph, Line 1 - Sentence "Access to cataract surgery..." how did you come to this conclusion

We have added the following reference to support our claim: An L, Jan CL, Feng J, Wang Z, Zhan L, Xu X. Inequity in access: cataract surgery throughput of Chinese ophthalmologists from the China national eye care capacity and resource survey. Ophthalmic Epidemiology. 2020 Jan 2;27(1):29-38.

The results

Are the results influenced by the fact that a significant number of patients (17.8%) did not come for the follow-up examination. Explain this in the limitations of the study.

Thank you for your suggestion. We have added this in the limitations (Page 23, lines 8-9).

Page 10, First paragraph - Please do not share the percentage values that are higher than those listed in Table 1.

Thank you for pointing out this detail. We have revised the values to make sure they are in accordance with Table 1 (Page 10, lines 4-8).

Table 1. Marital status may be shortened to: married or not

Thank you for the suggestion. We have modified the table according to your suggestion.

Table 2. What is the significance of this Table?

The table shows the association between provinces/regions and society-wide (GDP) and individual (household income) levels of wealth. We feel it is essential to the aim of the paper.

Table 3. What is the significance of this Table or is everything important already said in the paragraph above?

The table shows the association between the key geographical regions and individual level of wealth (household income), demographic characteristics (age, gender), social information (health insurance) and clinical output (cataract surgery output). As China is known to have large disparities (economic-wise and health access-wise) between its key regions, we feel this table contains valuable information.

Discussion

Page 21, First paragraph - Why did you compare the results from the population over 60 years old in your research with the results of other studies that presented the results from the population over 45 or 50 years old?

We reported the results from the population over 60 years old in our research because cataract is an age-related disease and we relied on self-reported data for cataract surgery rate and visual acuity. By using age 60 and older we filtered out potential confounding factors (other factors that can affect vision) when doing the regression analysis. Other studies clinically measured visual acuity and is therefore less prone to this confounding effect.

Are there any other studies from around the world that show the incidence of cataract surgery? Why are your results lower? Suggestion: does your study reflect real results while earlier results are actually limited?

We have reported results from other studies (references 8-12). Both the Beijing and the Liwan studies reported 5-year incidence where we reported 4-year incidence. That could explain why our incidence rate is a bit lower. However, due to study limitation (it was followed up in 4 year period rather than 5 year period), we cannot change the follow up period. We have added the follow up period of both the Beijing study and the Liwan study in our Discussion section to inform the readers.

Page 22, Second paragraph - The results that people who are illiterate, and on the other hand that people with higher incomes have a higher frequency of cataract surgery, are disproportionate. In addition, take into account the smaller number of operations in rural areas. The next page states that better economic status has a higher incidence of cataracts. This needs clarification! Is there a chance

that illiterate people gave wrong information about cataract surgery, and that they had other eye-related procedures?

I think Table 6 shows lower income is associated with less chance of cataract surgery. Regarding illiteracy, it COULD be confounded by inaccurate history, however it is very unlikely, because this survey was delivered by verbal interviews so misunderstanding even in an illiterate person is unlikely. As we have tried to explain in our discussion, it could be that the illiterate person is more likely to follow the doctor's recommendation rather than read it up on his own (e.g., Googling or the Chinese equivalent).

It can be concluded that people who have cataracts have a higher chance of having depression. Could it also be the reverse case - that some drugs accelerate the cataract development?

We are unaware of any evidence that depression drugs cause cataract. We have looked at PubMed and could not find any evidence that depression drugs cause cataract.

In the discussion, it is mentioned in several places that the implemented programs have improved the situation related to the availability of cataract surgery. On what basis can we come to that conclusion, which comparison was made and with what results? On the other hand, the results of this research do not show that? Additionally, could you explain is there a sufficient number of doctors (we have data), is there enough equipment (perhaps you have data) or is the insufficient patient education the biggest problem?

As a conclusion, are "great efforts" still needed to meet the needs for cataract surgery of the growing population? What should we pay attention to?

Thank you for the good questions. Unfortunately, this dataset cannot address these questions. We have modified the conclusion so that now it only addresses our aims. Now the conclusion reads: "Our study provides unique data on cataract surgical incidence in China. Our results suggest that surgical rates remain low. Rural and metropolitan both have access to cataract surgery, but underserved groups such as rural dwellers are less likely to receive cataract surgery."

References

The cited references are adequate, with a note that there are a couple of other studies (although they are from other populations) that could be included in the discussion of this paper.

Thank you for your comment.

VERSION 2 – REVIEW

REVIEWER	Jasmin Zvorničanin University Clinical Center Tuzla, Ophthalmology
REVIEW RETURNED	12-Feb-2023
GENERAL COMMENTS	Dear authors, For the most part, I am satisfied with the answers to the questions I have raised. However, I still have a few things I'd like to clarify. Before that, I would like to adequately answer the questions that have already asked. 1. The previous question was: „Methods - it is not clear from the methods presented in the paper which “clinical characteristics” were used. Does it refer to ophthalmic or systemic characteristics?” It I actually understand what you wanted to say. However, for our future readers, it is not clear when we say clinical characteristics. Please clarify this part? Proposal: "...measured biological, clinical, ophthalmic and socioeconomical characteristics..."

	2. The previous question was: „Why did you compare the results from the population over 60 years old in your research with the results of other studies that presented the results from the population over 45 or 50 years? “ You used the results of this research for people over 60 years old and compared them with earlier studies that presented the results in populations over 45 or 50 years. Why don't you compare your results with people over 45 and then list the potential limiting factors? 3. The previous question was: „Are there any other studies from around the world that show the incidence of cataract surgery? Why are your results lower? Suggestion: does your study reflect real results while earlier results are actually limited? “ I think that in the discussion you could make a better and more detailed comparison of your results with earlier research? 4. The previous question was: „Page 22, Second paragraph - The results that people who are illiterate, and on the other hand that people with higher incomes have a higher frequency of cataract surgery, are disproportionate. In addition, take into account the smaller number of operations in rural areas. The next page states that better economic status has a higher incidence of cataracts. This needs clarification! Is there a chance that illiterate people gave wrong information about cataract surgery, and that they had other eye- related procedures? “ In the Discussion Page 22 line 17 "Higher economic status (represented by higher...)" it is clearly stated that higher economic status is associated with a higher frequency of cataracts. I consider this interpretation of the results contradictory. I also consider the explanation "illiterate people were more likely to do what the doctor recommended" insufficient in this matter. Embed your explanation or comment in the paper itself. 5. The previous question was: „It can be concluded that people who have cataracts have a higher chance of having depression. Could it also be the reverse case - that some drugs accelerate the cataract development? „ It is stated in the text Page 11 Line 7: "higher depression scores at baseline" and also the results of Table 1 show that people with depression have earlier cataract surgery? Can you explain this in the context of your research results? Embed your explanation or comment in the paper itself.
--	--

VERSION 2 – AUTHOR RESPONSE

Reviewer: 2

Dr. Jasmin Zvorničanin, University Clinical Center Tuzla Comments to the Author:

Dear authors,

For the most part, I am satisfied with the answers to the questions I have raised. However, I still have a few things I'd like to clarify. Before that, I would like to adequately answer the questions that have already asked.

1. The previous question was: „Methods - it is not clear from the methods presented in the paper which

“clinical characteristics” were used. Does it refer to ophthalmic or systemic characteristics?”

It I actually understand what you wanted to say. However, for our future readers, it is not clear when we say clinical characteristics. Please clarify this part? Proposal: "...measured biological, clinical, ophthalmic and socioeconomical characteristics..."

Thank you for your suggestion. We have followed your proposal and provided further explanation for “clinical characteristics”. Specifically, we have added in the Introduction section just before the Method section: “The aim was to investigate incident cataract surgery, and to investigate the clinical (such as the presence of hypertension, diabetes, and depression).....” (Page 6 lines 11-12).

2. The previous question was: „Why did you compare the results from the population over 60 years old in your research with the results of other studies that presented the results from the population over 45 or 50 years? “ You used the results of this research for people over 60 years old and compared them with earlier studies that presented the results in populations over 45 or 50 years. Why don't you compare your results with people over 45 and then list the potential limiting factors?

Thank you for your helpful suggestion. We have followed your suggestion and compared our results with people over 45 in our Discussion section. The following is the edited text in our manuscript: “The current study found that between 2011 and 2015, the 4-year incidence of self-reported cataract surgery among people aged 45 and above was relatively low (1.22%) compared to earlier reports, which ranged from 2.9% (5-year incidence) in the Beijing Eye Study¹⁰ to 4.4% (5-year incidence) in Liwan, southern China.¹²” (Page 21, lines 5-7).

3. The previous question was: „Are there any other studies from around the world that show the incidence of cataract surgery? Why are your results lower? Suggestion: does your study reflect real results while earlier results are actually limited? “

I think that in the discussion you could make a better and more detailed comparison of your results with earlier research?

Thank you for your comment. We have followed your suggestion and have added more details discussing sample selection, definition of cataract surgery, and period of follow-up, into the Discussion section. The following texts have been added to the manuscript: “The current study found that between 2011 and 2015, the 4-year incidence of self-reported cataract surgery among people aged 45 and above was relatively low (1.22%) compared to earlier reports, which ranged from 2.9% (5-year incidence) in the Beijing Eye Study¹⁰ to 4.4% (5-year incidence) in Liwan, southern China.¹² It is worth noting that the Beijing and Liwan studies reported 5-year incidence rates, whereas our study reported 4-year incidence rate. This discrepancy may partly explain why our incidence rate appears to be lower. However, due to study limitations, we were unable to extend the follow-up period to 5 years. It is important to note that our sample was a nationally representative randomised sample selected from all provinces of China, from both urban and rural regions in each province, while previous studies only reported findings from one specific urban region in a particular province. This difference in sample selection may also contribute to the variance in incidence rates. Moreover, in our

study, cataract surgery was self-reported, whereas in other studies it was extracted from medical records or clinical examination.” (Page 21, lines 5-14)

4. The previous question was: „Page 22, Second paragraph - The results that people who are illiterate, and on the other hand that people with higher incomes have a higher frequency of cataract surgery, are disproportionate. In addition, take into account the smaller number of operations in rural areas. The next page states that better economic status has a higher incidence of cataracts. This needs clarification! Is there a chance that illiterate people gave wrong information about cataract surgery, and that they had other eye-related procedures? “

In the Discussion Page 22 line 17 "Higher economic status (represented by higher...)" it is clearly stated that higher economic status is associated with a higher frequency of cataracts. I consider this interpretation of the results contradictory. I also consider the explanation "illiterate people were more likely to do what the doctor recommended" insufficient in this matter. Embed your explanation or comment in the paper itself.

Thank you for your comment. We agree that this section needs more clarification. We have now incorporated your suggestion into the manuscript. The following changes have been made to the manuscript: “A noteworthy individual-level finding was a higher prevalence of self-reported illiteracy among those who underwent incident cataract surgery. Previous population studies have consistently linked poor education and lack of knowledge about cataract with low surgical uptake.²²⁻²³ One possible explanation for this finding is that illiterate people were more likely to comply with the doctor’s recommendations. Conversely, those with higher levels of education may have been more cautious about using local services and may have been unable to afford distant care. Additionally, those with lower literacy levels may have other unmeasured factors that predispose them to higher rates of cataract, such as greater exposure to UV light. It is also possible that illiterate individuals provided inaccurate information regarding their cataract surgery or had other eye related procedures, which resulted in relatively unreliable data concerning cataract surgery.” (Page 22, Lines 15-24)

5. The previous question was: „It can be concluded that people who have cataracts have a higher chance of having depression. Could it also be the reverse case - that some drugs accelerate the cataract development?

”

It is stated in the text Page 11 Line 7: "higher depression scores at baseline" and also the results of Table 1 show that people with depression have earlier cataract surgery? Can you explain this in the context of your research results? Embed your explanation or comment in the paper itself.

Thank you for your comment. We have incorporated the following text into the manuscript: “Additionally, cataract has been linked to higher rates of mental health problems, including depression,²⁹ which we believe could account for the observed higher rates of depression among those undergoing incident cataract surgery. To our knowledge, there is no evidence suggesting that depression drugs cause cataract. We conducted a thorough search of PubMed and were unable to find any studies supporting a causal link between depression drugs and cataract formation.” (Page 22, lines 11-15)

We hope that you find these changes satisfactory and that they have addressed your concerns. We believe that your suggestions have significantly improved the quality of our manuscript and we are grateful for your valuable feedback.

Once again, we would like to express our appreciation for your constructive comments and your time in reviewing our manuscript. We look forward to hearing from you again soon.

VERSION 3 – REVIEW

REVIEWER	Jasmin Zvorničanin University Clinical Center Tuzla, Ophthalmology
REVIEW RETURNED	24-Mar-2023

GENERAL COMMENTS	I am satisfied with the changes made. The authors also answered some of my additional questions that I had in mind. Therefore, I congratulate the authors on their excellent work and my recommendation is that this work should be accepted for publication in its current form.
---